# Gating of reafference in the external cuneate nucleus during self-generated movements in wake but not sleep

Alexandre Tiriac[1,2†], Mark S Blumberg[1,2,3*]

[1]Department of Psychological and Brain Sciences, The University of Iowa, Iowa City, United States; [2]The DeLTA Center, The University of Iowa, Iowa City, United States; [3]Department of Biology, The University of Iowa, Iowa City, United States

**Abstract** Nervous systems distinguish between self- and other-generated movements by monitoring discrepancies between planned and performed actions. To do so, corollary discharges are conveyed to sensory areas and gate expected reafference. Such gating is observed in neonatal rats during wake-related movements. In contrast, twitches, which are self-generated movements produced during active (or REM) sleep, differ from wake movements in that they reliably trigger robust neural activity. Accordingly, we hypothesized that the gating actions of corollary discharge are absent during twitching. Here, we identify the external cuneate nucleus (ECN), which processes sensory input from the forelimbs, as a site of movement-dependent sensory gating during wake. Whereas pharmacological disinhibition of the ECN unmasked wake-related reafference, twitch-related reafference was unaffected. This is the first demonstration of a neural comparator that is differentially engaged depending on the kind of movement produced. This mechanism explains how twitches, although self-generated, trigger abundant reafferent activation of sensorimotor circuits in the developing brain.

*For correspondence: mark-blumberg@uiowa.edu

Present address: †Department of Molecular and Cell Biology, University of California, Berkeley, Berkeley, United States

**Competing interests:** The authors declare that no competing interests exist.

## Introduction

Animals of diverse vertebrate and invertebrate species distinguish between sensations arising from self-generated movements from those arising from other-generated movements (*Crapse and Sommer, 2008*; *Sommer and Wurtz, 2008*). To make this distinction between self and other, motor areas produce copies of motor commands (i.e., corollary discharges) that are directly compared with sensory signals arising from self-generated movements (i.e., reafference; [*Poulet and Hedwig 2006*; *Sommer and Wurtz 2002*]). At the level of the neural comparator, reafference is blocked when there are no discrepancies between the corollary discharge and reafferent signals (*Poulet and Hedwig 2006*; *Bell, 2001*; *Brooks and Cullen, 2014*; *Bell, 1989*). As a result, expected reafference arising from voluntary movements is attenuated (*Voss et al., 2006*; *Seki et al., 2003*; *Haggard and Whitford 2004*), thereby increasing the salience of unexpected reafference (*Poulet and Hedwig 2006*; *Brooks et al., 2015*; *Brooks and Cullen, 2013*).

We previously demonstrated in week-old rats that wake-related movements do not trigger substantial reafference in sensorimotor cortex (SMC). In contrast, sleep-related twitches—which are produced exclusively and abundantly during active (or REM) sleep—trigger robust reafference (*Tiriac et al., 2014*). We provided converging evidence that twitches, unlike wake movements, are processed by the nervous system *as if* they are unexpected. Accordingly, we hypothesized that the actions of corollary discharge that normally gate reafference arising from wake movements are either absent or inhibited during twitching. If true, there must be a neural structure, located somewhere

**eLife digest** Many parts of our body twitch while we are asleep and these movements are especially common in babies. Unlike the movements we make while awake, twitches during sleep are brief, staccato-like movements that appear to be aimless – but they are not, as traditionally believed, mere remnants of dreams. Rather, it is believed that twitches help infants learn about their bodies and how they move.

When we are awake, the brain routinely compares information from the areas of the brain that produce movements with information coming in from the senses, so that we are better able to anticipate and control the movements we make. However, this comparison appears to be suspended during twitching: in 2014, researchers studying infant rats reported that twitches, although self-produced, are treated by the brain as if they are unexpected or surprising.

To confirm that the brain treats twitches differently from wake movements, Tiriac and Blumberg – who were involved in the previous study – recorded electrical activity in the brains of infant rats while they were awake and asleep. These experiments show that a brain area known as the external cuneate nucleus (ECN) was mostly inactive when awake rats vigorously moved their front legs, but became highly active when these same legs twitched during sleep. Drugs that disinhibited electrical activity in the ECN unmasked leg movement signals produced by awake rats, but these same drugs had no effect on leg movement signals produced during twitching. Thus, these experiments indicate that, when infant rats are awake, the ECN compares signals from the senses with signals from the parts of the brain that produce movements, a key feature of motor control. However, when the rats are asleep and twitching, the comparison mechanism is disengaged and sensory signals are allowed to cascade through the ECN to many other structures in the brain.

Tiriac and Blumberg's findings open new avenues for understanding how the developing brain learns to distinguish the movements that we produce ourselves from those that occur due to forces in the outside world. Also, the challenge remains to identify the specific mechanisms by which twitches help develop and refine the brain circuits that enable mammals to move around as effectively as they do.

within the sensorimotor network, that acts as a comparator to specifically gate wake-related reafference.

One possible comparator is the external cuneate nucleus (ECN), which receives primary muscle spindle afferents from forelimb and nuchal muscles and conveys sensory information to downstream structures including the cerebellum, thalamus, and cerebral cortex (*Campbell et al., 1974*; *Cooke et al., 1971*; *Mackie et al., 1999*; *Boivie and Boman, 1981*; *Huang et al., 2013*). Moreover, the ECN receives projections from such premotor structures as the red nucleus (*Holstege and Tan, 1988*; *Edwards, 1972*; *Martin et al., 1974*)—which contributes both to the production of wake movements and twitches in infant rats (*Del Rio-Bermudez et al., 2015*)—and the premotor C3-C4 propriospinal neurons (PNs; [*Pivetta et al., 2014*]), which have been implicated in the conveyance of corollary discharge (*Pivetta et al., 2014*; *Alstermark et al., 2007*; *Azim et al., 2014*). Corollary discharge from the red nucleus or PNs could contribute to sensory gating in the ECN via known GABAergic and glycinergic inputs to that structure (*Galindo and Krnjević, 1967*; *Heino and Westman 1991*; *Sato et al., 1991*), perhaps through presynaptic inhibition (*Seki et al., 2003*; *Andersen et al., 1964*).

## Results

### Twitch-related, but not wake-related, movements trigger reafference in SMC

We first recorded neural activity in the forelimb region of SMC to confirm that it, like the hindlimb region (*Tiriac et al., 2014*), exhibits state-dependent activity. Unanesthetized and head-fixed postnatal day (P) 8–10 rats cycled spontaneously between sleep and wake with their limbs dangling freely. *Figure 1A* depicts representative spindle bursts (recorded from the local field potential; LFP)

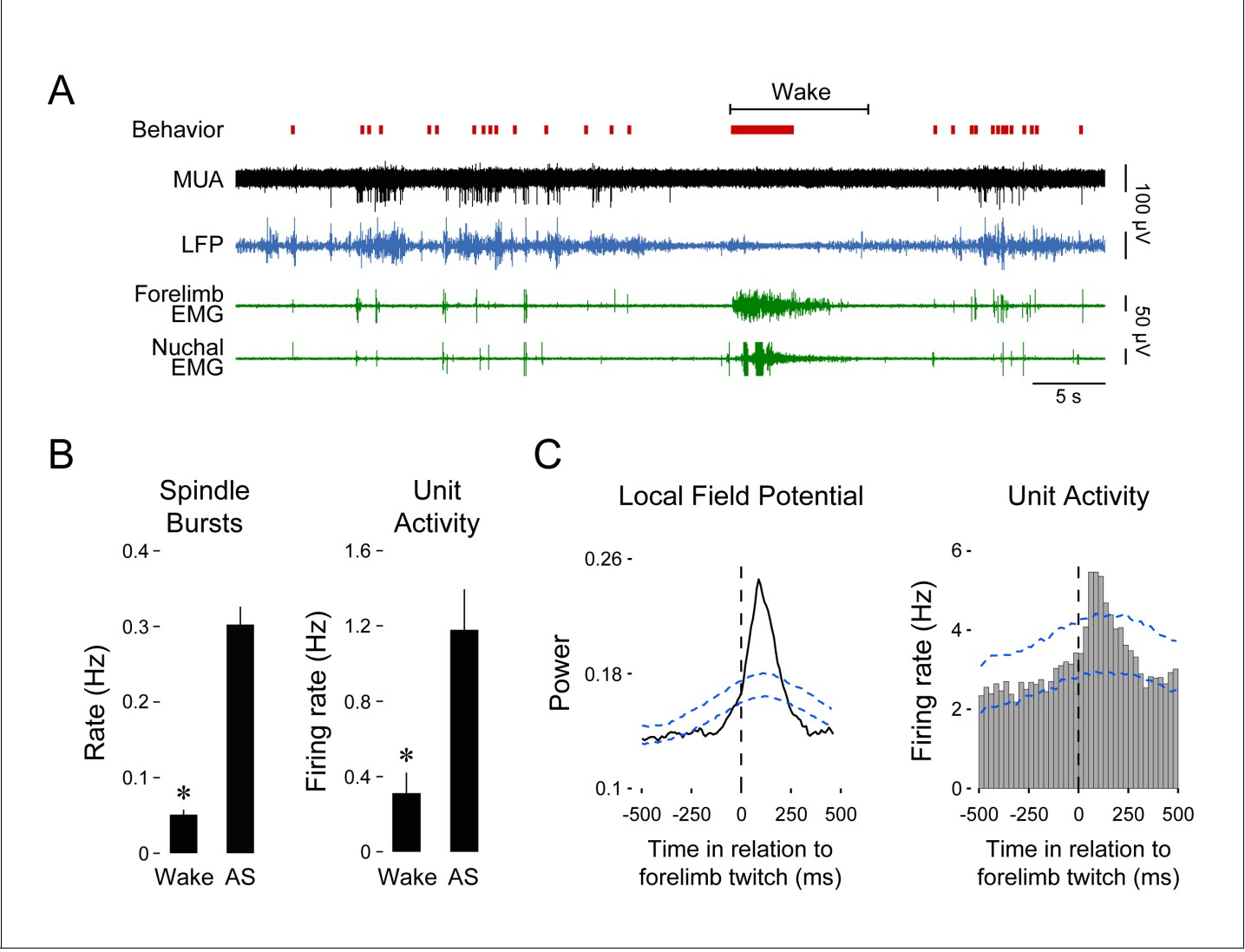

**Figure 1.** Forelimb twitches, but not wake movements, trigger neural activity in forelimb sensorimotor cortex. (**A**) Representative data depicting sleep and wake behavior, MUA, LFP, and forelimb and nuchal EMG during spontaneous sleep-wake cycling. Red tick marks denote forelimb twitches and red horizontal bars denote forelimb wake movements as scored by the experimenter. (**B**) Mean (+SEM) rate of spindle burst (n = 6 pups) and unit activity (n = 8 units) during periods of wake and active sleep (AS). The mean rate of spindle bursts and unit activity was significantly higher during active sleep than during wake. * significant difference from active sleep, p<0.01. (**C**) Waveform average and event correlation for LFP power and unit activity, respectively, in relation to forelimb twitches (2413 and 2943 twitches, respectively). The blue dashed lines denote upper and lower acceptance bands (p≤0.01). LFP, Local field potential; MUA, Multiunit activity.

and multiunit activity (MUA) in relation to both wake movements and twitches. Spindle bursts and MUA were particularly prominent during periods of twitching and were virtually absent during periods of wake movements. Across all pups, the mean rate of spindle bursts ($t_5$ = 10.2, p=0.0002, n = 6 pups) and unit activity ($t_7$ = 3.9, p=0.006, n = 8 units) was significantly higher during active sleep than during wake (*Figure 1B*). Moreover, spindle bursts and unit activity were triggered by forelimb twitches with a latency of 100–125 ms (*Figure 1C*).

## Twitch-related, but not wake-related, movements trigger reafference in the ECN

If the ECN is a comparator of reafference and corollary discharge signals, then it should—like the SMC—exhibit state-dependent activity. Therefore, we next recorded from the ECN in P8-10 rats

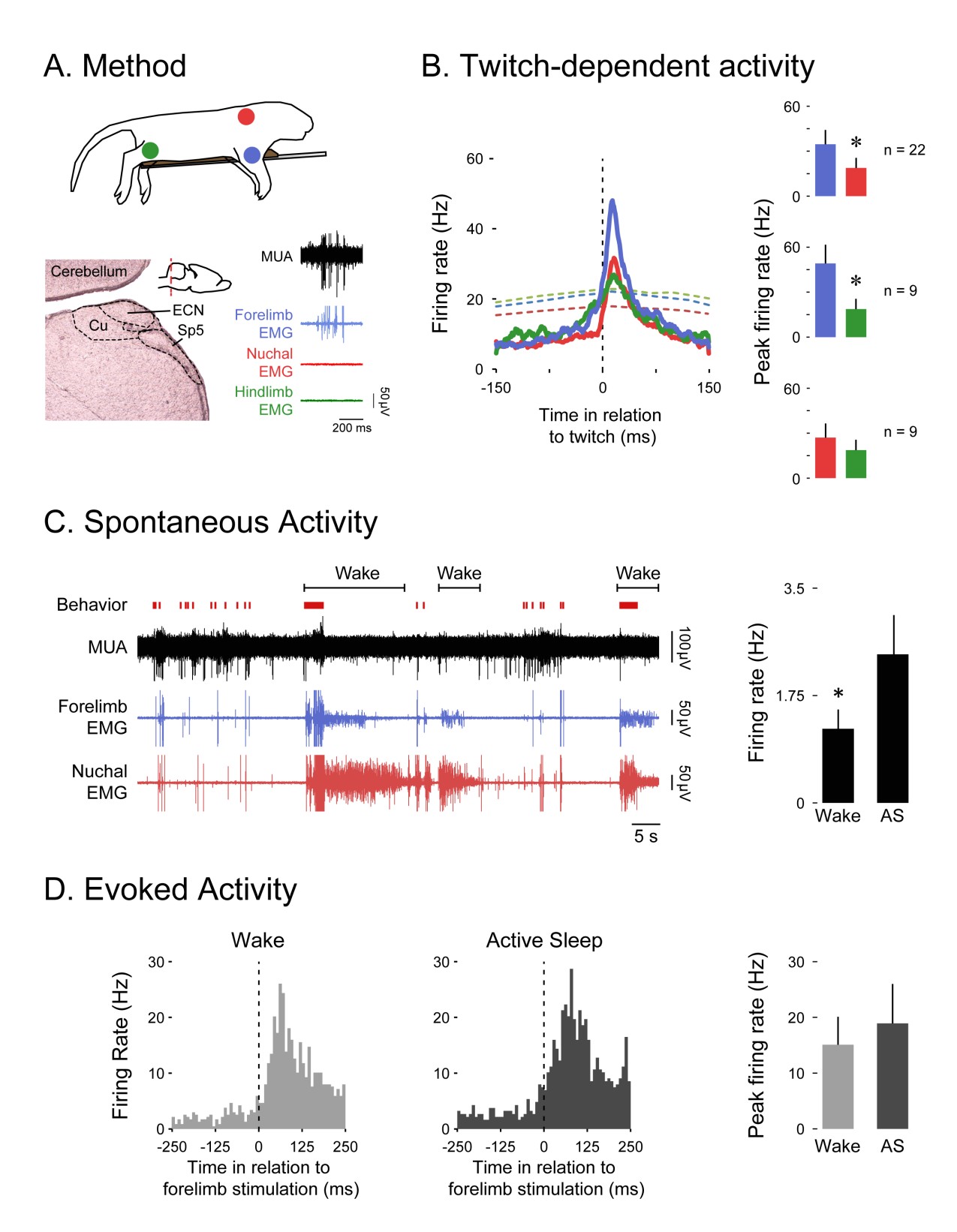

**Figure 2.** The ECN exhibits wake-dependent inhibition of sensory reafference. (**A**) Top: For all ECN recordings, P8-10 rats were instrumented with forelimb (blue) and nuchal (red) EMGs (n = 22). A subset of these rats also had a hindlimb (green) EMG (n = 9). The torso was supported by a platform

*Figure 2 continued on next page*

*Figure 2 continued*

and the limbs dangled freely. Bottom left: Coronal brain section depicting the anatomical location of the ECN in the hindbrain (inset; red dashed line depicts AP position of the coronal section). Bottom right: Sample record of a burst of ECN reafference in response to forelimb twitches. (B) Left: Event correlations for unit activity in relation to forelimb (blue, n = 8146 twitches), nuchal (red, n = 9603 twitches), and hindlimb (green, n = 1243 twitches) twitches. The colored dashed lines denote upper acceptance bands (p≤0.01) for the event correlations. Right: Pairwise comparisons of mean (+SEM) peaks in unit activity (Hz) in response to forelimb, nuchal, and hindlimb twitches. Comparisons are between forelimb and nuchal muscles (top), forelimb and hindlimb muscles (middle), and nuchal and hindlimb muscles (bottom). (C) Left: Representative data depicting sleep and wake behavior, MUA, and forelimb and nuchal EMG during spontaneous sleep-wake cycling. Red tick marks denote forelimb twitches, and red horizontal bars denote forelimb wake movements as scored by the experimenter. Right: Mean (+SEM) unit activity (n = 16) during wake and active sleep (AS) periods. * significant difference from active sleep, p<0.05. (D) Left: Event correlations for evoked unit activity in response to forelimb stimulations performed during active sleep (n = 188 stimulations) and wake (n = 238 stimulations) across 6 ECN units in six pups. Right: Mean (+SEM) peak unit activity (n = 6) derived from event correlations during wake and active sleep. ECN, External cuneate nucleus; MUA, Multiunit activity.

(*Figure 2A*). As expected, ECN firing rate was higher in response to forelimb twitches than to hindlimb ($t_8$ = 3.5, p=0.008) or nuchal ($t_{21}$ = 3.2, p=0.007) twitches (*Figure 2B*); there was no significant difference in ECN firing rate between hindlimb and nuchal twitches ($t_8$ = 1.3, p=0.23). Also, as expected given the relative anatomical locations of the ECN and SMC, the latency to twitch-related ECN activity (10–50 ms) was shorter than that observed in SMC (100–125 ms; see *Figure 1C*). Finally, ECN firing rate was movement dependent, with greater activity occurring during periods of forelimb twitching than during periods of forelimb wake movement ($t_{15}$ = 2.8, p=0.012, *Figure 2C*).

It is possible that the inhibition of wake-related reafference was due to a global inhibition of all sensory activity during wake. If so, then stimulation of the ipsilateral forelimb should produce less exafference in the ECN during wake than during sleep. To test for this possibility, we recorded from ECN neurons as the ipsilateral forelimb was stimulated during sleep and wake. As shown in *Figure 2D*, there was no significant effect of behavioral state on ECN activity in response to forelimb stimulation ($t_5$ = 1.2, p=0.286). This result is consistent with our earlier finding in the hindlimb region of SMC (*Tiriac et al., 2014*).

## Disinhibition of the ECN unmasks wake-related reafference

*Figure 3A* depicts our proposed model to explain movement-dependent modulation of reafference in the ECN. To test the model's predictions, we recorded ECN activity before and during combined iontophoretic infusion of $GABA_A$ (10 mM bicuculline methiodide) and glycine (10 mM strychnine hydrochloride) receptor antagonists or saline (*Figure 3B*; n = 5 pups per group). As predicted, inhibitory blockade of the ECN unmasked reafference in response to forelimb wake movements ($t_8$ = 3.7 p=0.006, *Figure 3C*). Moreover, and again as predicted, inhibitory blockade had no effect on ECN reafference in response to forelimb twitches (*Figure 3D*). Saline infusions had no effect on either wake or sleep reafference.

We next assessed whether inhibitory blockade of the ECN alters motor activity. Blocking $GABA_A$ and glycine receptors had no effect on the amplitude of wake movements (*Figure 4A*), the frequency of forelimb wake movements (*Figure 4B*), or the frequency of forelimb twitches (*Figure 4C*). Moreover, inhibitory blockade had no effect on tonic neural activity in the ECN (*Figure 4D*), thus providing further evidence that the inhibitory inputs to the ECN function specifically in the context of wake movements.

## Discussion

### The ECN is a neural comparator

To establish that the ECN is a comparator of corollary discharge and reafferent signals, several criteria must be met (*Poulet and Hedwig, 2007*). First, to be a comparator, the ECN must be a sensory structure; in fact, the ECN processes proprioceptive inputs from forelimb and nuchal muscles (*Campbell et al., 1974*) and recordings from anesthetized cats have demonstrated that the ECN codes, with high fidelity, the stretch on muscle fibers (*Mackie et al., 1999*). Second, the ECN must receive motor-related input; in fact, the ECN receives direct and indirect input from premotor areas, including the red nucleus (*Holstege and Tan, 1988*; *Edwards, 1972*; *Martin et al., 1974*) and C3-

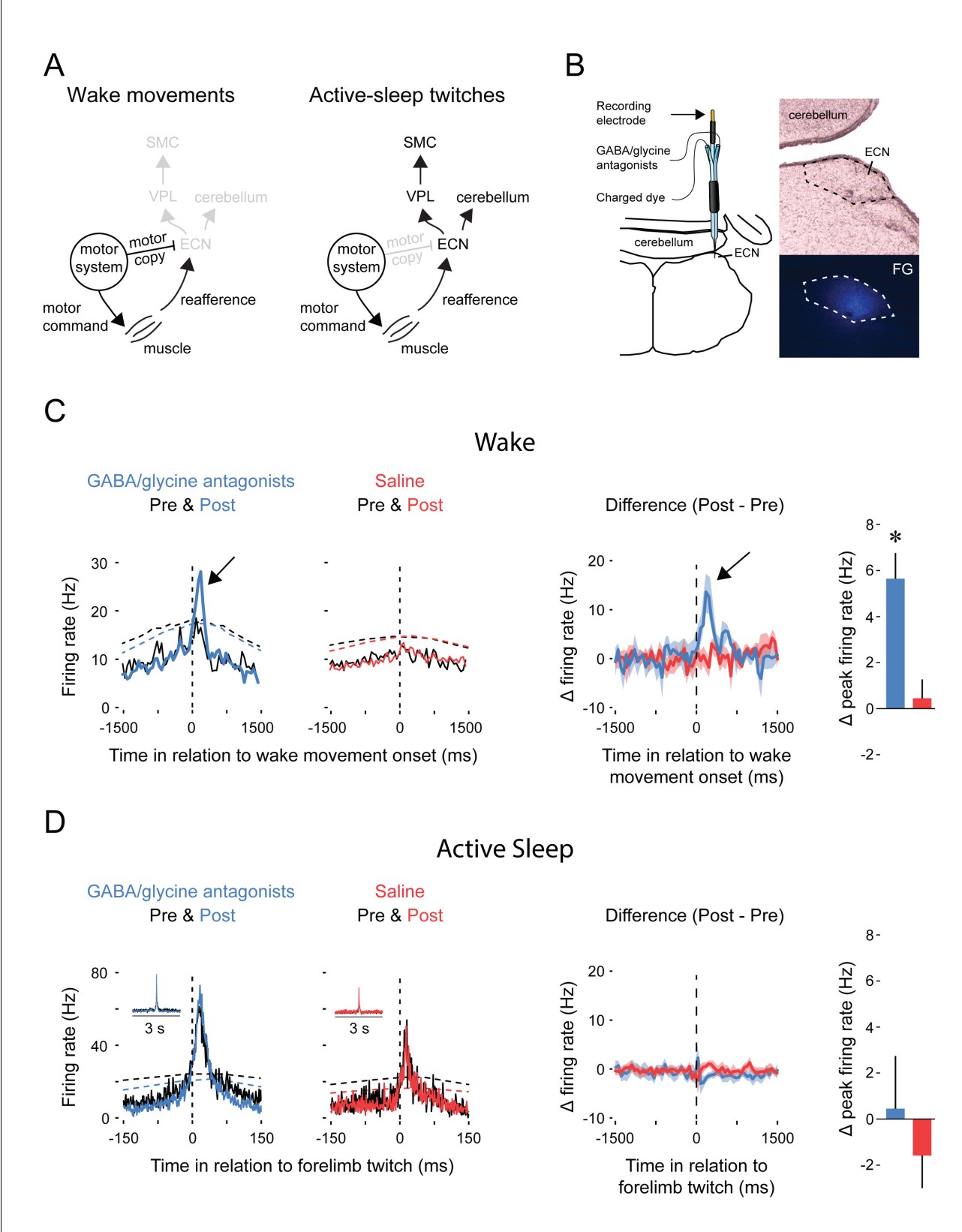

**Figure 3.** Pharmacological blockade of GABA$_A$ and glycine receptors in the ECN specifically unmasks reafference from self-generated wake movements. (**A**) Proposed circuitry depicting how reafference arising from self-generated movements is modulated at the level of the ECN. Left: During

*Figure 3 continued on next page*

*Figure 3 continued*

wake, ECN inputs arising from motor areas convey a corollary discharge (i.e., motor copy) signal that gates expected reafference, resulting in decreased activity in the ECN and downstream sensory areas. Right: During active sleep, the motor copy is absent or inhibited and, therefore, ECN inputs arising from motor areas do not gate reafference, resulting in activity in the ECN and downstream sensory areas. VPL: ventral posterolateral nucleus of thalamus; SMC: primary sensorimotor cortex. (B) ECN recordings were performed using multibarrel electrodes filled either with a GABA$_A$ antagonist (10 mM bicuculline methiodide) and a glycine antagonist (10 mM strychnine hydrochloride) or saline. For all animals, a separate barrel of the electrode was filled with fluorogold (FG) to mark the location of the recording site and estimate the spread of the drug (image at bottom right). (C) Left: Event correlations for unit activity in relation to the onset of forelimb wake movements in animals in the GABA/glycine (blue) or saline (red) groups before (Pre) and after (Post) infusion. Data are pooled across all pups. The dashed lines denote upper acceptance bands (p≤0.01) for the event correlations. Right: Event correlations depicting changes in unit activity between the pre-infusion and post-infusion periods for the GABA/glycine (blue) and saline (red) groups. Color-coded shaded regions denote ±SEM. Histograms depict mean (+SEM) peak changes in unit activity. n = 5 per group. * significant difference from saline, p<0.05. (D) Same as in (C) but during active sleep; event correlations are triggered on forelimb twitches. ECN, External cuneate nucleus.

C4 PNs (*Pivetta et al., 2014*). Third, the ECN should not participate in the production of movement; as shown here, disinhibition of the ECN had no discernible effect on the production of wake movements or twitches (*Figure 4*).

Finally, and most critically, a comparator must gate reafference arising from movements. This was demonstrated here by showing that inhibitory blockade of the ECN unmasked reafference exclusively during wake movements (*Figure 3C*). Importantly, inhibitory blockade had no effect on the ECN's tonic firing rate (*Figure 4D*), thus demonstrating that inhibitory control of the ECN is not engaged throughout the waking state. Furthermore, when we manually stimulated the forelimbs, ECN activity was evoked similarly during sleep and wake (*Figure 2D*).

## Possible mechanisms underlying movement-dependent modulation of reafference

There is no evidence that wake movements and twitches are produced by independent brainstem structures. On the contrary, the red nucleus is involved in the production of both types of movements (*Del Rio-Bermudez et al., 2015*; *Gassel et al., 1966*). Therefore, it may be that (a) premotor structures like the red nucleus are state-dependently modulated such that they convey motor copies to the ECN (or an intervening structure) only during wake movements, or (b) the ECN receives motor copies from both types of movements, but state-dependent modulation of the ECN prevents the twitch-related motor copies from engaging the inhibitory mechanism. Indeed, in neonatal rats, there are structures, including the noradrenergic locus coeruleus, that could play a state-dependent modulatory role (*Karlsson et al., 2005*). Regardless of which mechanism turns out to be correct, the fact remains that wake movements and twitches are processed very differently at the level of the ECN.

## Functional implications

Experimental disruption of corollary discharge pathways has a negative impact on forelimb reaching (*Azim et al., 2014*) and control of eye movements (*Sommer and Wurtz 2002*). In this context, the gating of wake-related reafference within the ECN makes functional sense. But if sensory gating is critical for motor function, why disengage this mechanism during sleep-related twitches?

The absence of sensory gating during sleep raises the possibility that corollary discharge mechanisms interfere with functional processes in which twitches are involved. Indeed, if sensory gating were to occur during active sleep, twitches—which are especially abundant during early development—would trigger little or no reafference. In the context of a developing system that relies heavily on activity-dependent processes (*Kirkby et al., 2013*), the gating of twitch-related reafference would suppress the very activity upon which some of these processes depend. Because reafference is not gated during twitching, it is permitted to sequentially activate interconnected sensorimotor structures—including the ECN (as shown here), red nucleus (*Del Rio-Bermudez et al., 2015*), cerebellum (*Sokoloff et al., 2015a*; *2015b*), thalamus (*Tiriac et al., 2012*; *Khazipov et al., 2004*), sensorimotor cortex (*Tiriac et al., 2014*; *Khazipov et al., 2004*), and hippocampus (*Mohns and Blumberg, 2010*). Such cascading neural activity provides the opportunity for competitive synaptic interactions that, through such mechanisms as spike-timing-dependent plasticity (*Feldman, 2012*;

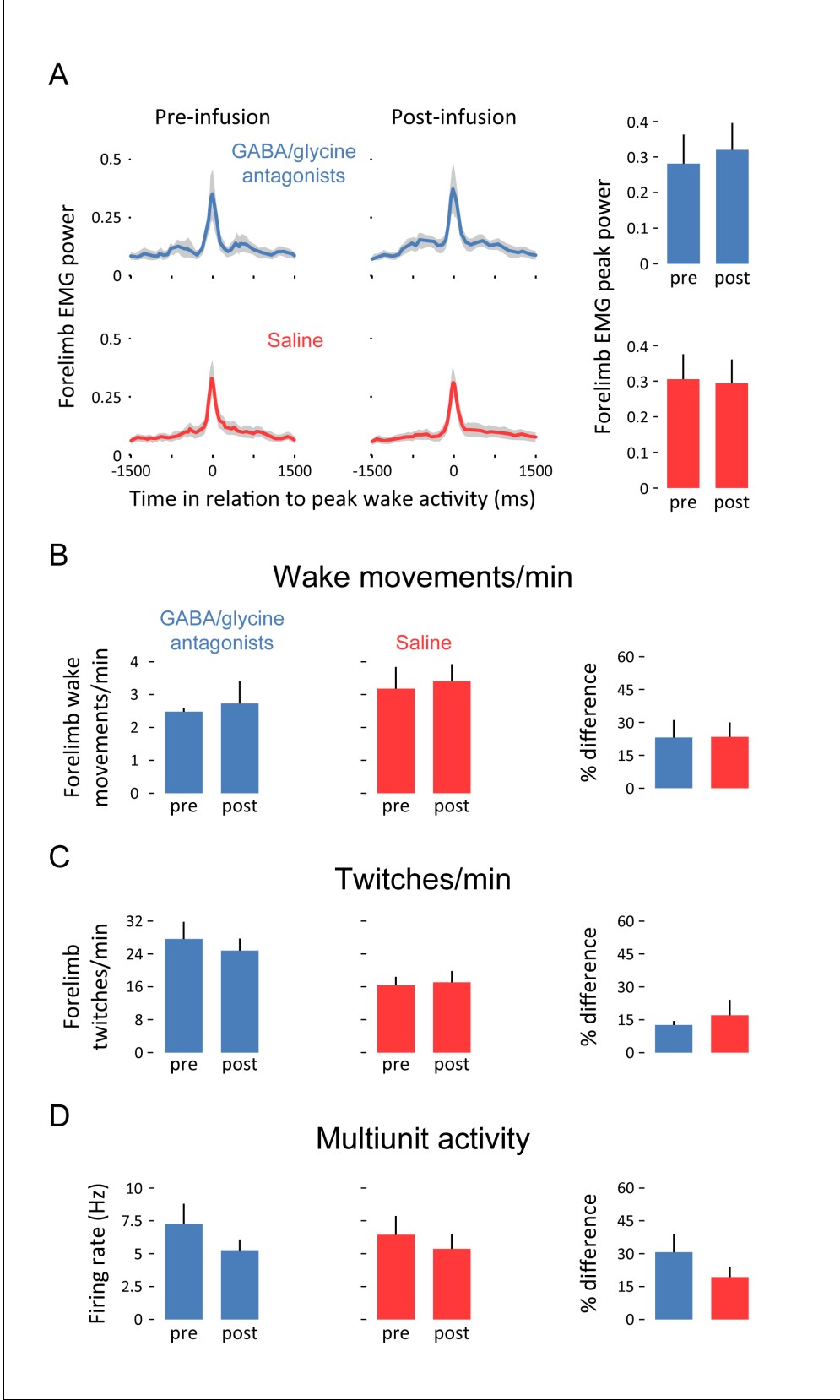

**Figure 4.** Pharmacological blockade of GABA$_A$ and glycine receptors in the ECN does not affect forelimb motor activity or tonic ECN unit activity. (**A**) Waveform averages (3-s time windows) depicting forelimb EMG power in relation to peak forelimb wake activity before and after infusion of GABA$_A$
*Figure 4 continued on next page*

*Figure 4 continued*

and glycine receptor antagonists (blue) or saline (red). Shaded regions denote $\pm$SEM. At right, mean (+SEM) forelimb EMG peak power derived from waveform averages. (B) Top row: Mean (+SEM) forelimb wake movements/min before and after infusion of GABA$_A$ and glycine receptor antagonists (blue) or saline (red). At right, mean (+SEM) percent difference (pre vs. post) in forelimb wake movements/min for both experimental groups. (C) and (D): Same as in (B) but for twitches/min and unit activity (in Hz), respectively. None of the differences in this figure are significant. ECN, External cuneate nucleus.

*Song and Abbott, 2001*), can contribute to the development and refinement of somatotopic organization across the neuraxis.

The very presence of a neural comparator that is differentially engaged in a movement-dependent manner argues both for the importance of sensory gating during wake and its absence during sleep. With regard to the latter, the identification here of a neural mechanism that distinguishes twitches from wake movements reinforces the notion that twitches play a critical role in shaping sensorimotor circuits (*Blumberg et al., 2013*).

## Materials and methods

All experiments were carried out in accordance with the National Institutes of Health Guide for the Care and Use of Laboratory Animals (NIH Publication No. 80–23) and were approved by the Institutional Animal Care and Use Committee of the University of Iowa.

### Subjects

A total of 42 Sprague-Dawley Norway rats (RRID:RGD_5508397) were used at P8-10. Males and females were used and littermates were always assigned to different experimental groups. Litters were culled to eight pups within 3 days of birth. Mothers and their litters were housed in standard laboratory cages (48 $\times$ 20 $\times$ 26 cm). Food and water were available *ad libitum*. All animals were maintained on a 12:12 light-dark schedule with lights on at 0700 hr.

### Surgery

#### Head-fix preparation

For all studies, pups were prepared for testing using methods similar to those described previously (*Del Rio-Bermudez et al., 2015*; *Sokoloff et al., 2015a*; *Tiriac et al., 2012*; *Blumberg et al., 2013*; *Blumberg et al., 2015*). Under isoflurane anesthesia, bipolar electrodes (50 µm diameter; California Fine Wire, Grover Beach, CA) were implanted into the *biceps brachii* muscle of the forelimb, the *extensor digitorum longus* muscle of the hindlimb, and the nuchal muscle. The skin overlying the skull was removed and a custom-built head-fix apparatus was attached to the skull with cyanoacrylate adhesive. A small hole was drilled over the forelimb region of primary sensorimotor cortex (SMC). Forelimb, coordinates in relation to bregma: AP: 1 mm, L: 1.3–1.8 mm, DV: 0.5–0.8 mm) or over the external cuneate nucleus (ECN, coordinates in relation to lambda: AP: −3 mm, L: 1.6 mm, DV: 3.5–4 mm, angle of electrode: 14° directed caudally). After surgery, the pup was transferred to a humidified incubator maintained at thermoneutrality (35°C) to recover for at least 1 hr, after which it was transferred to a stereotaxic apparatus. The pup's torso was supported on a narrow platform such that the limbs dangled freely on both sides. The pup acclimated for at least one additional hour before recordings began, by which time it was cycling between sleep and wake.

### Electrophysiological recordings

The electromyographic (EMG) electrodes were connected to a differential amplifier (A-M Systems, Carlsborg, WA; amplification: 10,000x; filter setting: 300–5000 Hz). To record from forelimb SMC, 16-channel silicon depth electrodes were used (100 µm vertical separation; NeuroNexus, Ann Arbor, MI). To record from the ECN, four-channel silicon depth electrodes were used (50 µm vertical separation). Silicon electrodes had impedances ranging from 1 to 4 MΩ. To simultaneous perform ECN recordings during iontophoretic application of GABA$_A$ and glycine receptor antagonists, multibarrel electrodes were used with an iontophoretic pump (Kation Scientific, Minneapolis, MN). Electrodes were connected to a headstage which communicated with a data acquisition system (Tucker-Davis

Technologies, Alachua, FL) that amplified (10,000x) and filtered the signals. All cortical recordings were obtained using a 5000 Hz low-pass filter, and all recordings in ECN were obtained using a 500–5000 Hz band-pass filter. A 60-Hz notch filter was also used. Neurophysiological and EMG signals were sampled at 25 kHz and 1 kHz, respectively, using a digital interface and Spike2 software (Cambridge Electronic Design, Cambridge, UK).

Prior to insertion of the silicon probe into SMC or ECN, the electrode surface was coated with fluorescent DiI (Life Technologies, Carlsbad, CA) for subsequent histological verification of electrode placement. A Ag/AgCl ground electrode (Medwire, Mt. Vernon, NY, 0.25 mm diameter) was placed into the visual cortex ipsilateral to the silicon probe. Brain temperature was monitored using a fine-wire thermocouple (Omega Engineering, Stamford, CT) placed in the visual cortex contralateral to the ground wire. For all experiments, brain temperature was maintained at 36–37 °C.

Electrode position was established when it was possible to reliably evoke neural activity by gentle stimulation of the contralateral (for sensorimotor cortex) or ipsilateral (for ECN) forelimb. Other parts of the body were also stimulated to confirm selectivity. Using procedures similar to those described previously (*Del Rio-Bermudez et al., 2015*; *Sokoloff et al., 2015a*; *Tiriac et al., 2012*), data acquisition began after local field potentials (LFP) and multiunit activity (MUA) were identified and had stabilized for at least 10 min.

## Experimental procedure

### Spontaneous activity during sleep and wake

We recorded spontaneous neural activity in cycling unanesthetized infant rats (n = 6 for SMC recordings, n = 14 for ECN recordings). Recording sessions comprised continuous collection of neurophysiological and EMG data for at least 30 min. During acquisition, an experimenter monitored the subject's behavior and digitally marked the occurrence of sleep-related twitching and wake movements in synchrony with the physiological data. For the scoring of limb movements, we used two sets of digital markers. One set of markers was used for twitches and wake movements of the forelimb, and the other set was used for twitches and wake movements of the other limbs. As described elsewhere (*Karlsson et al., 2005*), myoclonic twitches are phasic, rapid, and independent movements of the limbs and tail against a background of muscle atonia; in contrast, wake movements, which are high-amplitude, coordinated movements occurring against a background of high muscle tone, include kicking, stretching, and yawning. Finally, the experimenter was always blind to the physiological data when scoring behavior.

### Forelimb stimulation

In 6 of the 14 animals in which spontaneous sleep-wake activity was recorded in the ECN, the forelimb ipsilateral to the recording site was stimulated using a fine paintbrush over a period of 10 min. The brush was applied to the base of the paw and the limb was flexed at the elbow. The stimulations were performed similarly during sleep and wake. On average, stimulations were repeated every 5–10 s.

### Iontophoretic infusions of GABA$_A$ and glycine receptor antagonists

In 10 head-fixed P8-10 rats, ECN activity was recorded before and during iontophoretic infusions of GABA$_A$ (10 mM bicuculline methiodide) and glycine (10 mM strychnine hydrochloride; Sigma-Aldrich, St. Louis, MO) receptor antagonists or saline. Before infusion, 30 min of baseline ECN activity was recorded. During this recording, a −10 nA retaining current was used to ensure that the antagonists did not leak passively into the ECN. After the baseline recording, the current was switched to +50 nA to eject the drugs and another 30 min of ECN activity was recorded. After this recording period, fluorogold was infusedwith the same current parameter that was used for drug infusion to mark the location of the electrode.

## Histology

At the end of the recording session, the pup was overdosed with sodium pentobarbital (1.5 mg/g) and perfused transcardially with phosphate-buffered saline followed by 4% paraformaldehyde. Brains were sectioned at 50 μm using a freezing microtome (Leica Microsystems, Buffalo Grove, IL). Recording sites were verified by visualizing the DiI tract or fluorogold at 5–20X magnification using a

fluorescent Leica microscope. Tissue slices were then stained using cresyl violet and the location of the recording site was identified.

## Data analysis

### MUA and LFP analysis

All channels were filtered for multiunit activity (500–5000 Hz). Spike sorting was performed using Spike2 (Cambridge Electronic Design, Cambridge, UK). For LFPs, channels were band-pass filtered (1–40 Hz) to extract spindle bursts. Spindle bursts were defined as comprising at least three oscillations, a dominant frequency of 10–15 Hz, and a duration of at least 100 ms (*Khazipov et al., 2004*). LFP channels were processed using root mean square (RMS) with a time constant of 0.01 s. Five random high-amplitude spindle bursts were averaged and the baseline value of the RMS channel was calculated. The midpoint between those two values was used as a threshold for identification of spindle bursts. A second pass through the data was performed to manually remove any spindle bursts that did not match the requisite criteria. Throughout this process, artifacts in the LFP and MUA signals were identified and manually removed.

### Identification of behavioral state and motor activity

Sleep and wake periods were defined using methods described previously (*Karlsson et al., 2005*; *Tiriac et al., 2012*). Briefly, the nuchal EMG signal was dichotomized into periods of high tone (indicative of wake) and atonia (indicative of sleep). Active sleep was characterized by the occurrence of myoclonic twitches against a background of muscle atonia (*Seelke and Blumberg, 2008*). Spikes of EMG activity with amplitudes greater than 3x baseline were considered twitches.

For the identification of forelimb wake movements, the forelimb EMG was rectified and smoothed (0.01 ms). To be considered a wake movement, a spike in the forelimb EMG had to be at least 300 ms in duration and had to occur while nuchal muscle tone was high (indicative of wake). Five random wake movements that met these criteria were selected and their maximum amplitude was averaged. Using the midpoint between atonia and max amplitude of wake movements as a threshold, an automatic peak detection was applied to extract forelimb movements as events. Similar to spindle burst detection, a second pass through the data was performed to manually remove any wake movements that did not match the requisite criteria.

### Analysis of state dependency

There were at least 20 bouts of wake and active sleep for each pup. Across these bouts, the mean rates of spindle burst production and unit activity were determined. First, for each individual pup, successive bouts of wake and active sleep were treated as pairs and the Wilcoxon matched-pairs signed-ranks test (SPSS, IBM, Armonk, NY) was used to test for differences in rates of spindle burst production and unit activity between the two states. Second, the mean rates of spindle burst production and unit activity during wake and active sleep were calculated for each pup and compared within each age group using paired *t* tests.

### Event correlations and waveform averages for SMC and ECN activity triggered by twitches

The relationship between twitches and SMC and ECN activity was assessed as follows: First, data for all pups were concatenated into one file. From this file, using twitches as triggers, event correlations (for SMC: 1000-ms windows, 25-ms bins; for ECN: 300-ms windows, 1-ms bins) of unit activity and waveform averages of spindle activity were constructed. We tested statistical significance for both event correlations and waveform averages by jittering twitch events 1000 times within a 500-ms window using the interval jitter parameter settings within PatternJitter (*Amarasingham et al., 2012*; *Harrison, 2009*) implemented in Matlab (MathWorks, Natick, MA). We corrected for multiple comparisons using the method of *Amarasingham et al. (2012)*; this method produces upper and lower acceptance bands (p<0.01) for each event correlation and waveform average.

### Stimulus-triggered event correlations and waveform averages

The 10-min recordings from the forelimb stimulation trials were divided into periods of sleep and wake. Perievent histograms (500-ms windows, 10-ms bins) were constructed using onset of EMG

activity produced by forelimb stimulations as the trigger. Average peak firing rate across all stimulations for each behavioral state was determined. Peak firing rate was compared using a paired t test.

### Effects of GABA$_A$ and glycine receptor antagonists on ECN neural activity

For each animal in each group, recordings were divided into a pre- and post-infusion period (30 min each). For each time period and for each behavioral state, event correlations (3000-ms windows, 50-ms bins) were generated to correlate MUA activity to ipsilateral forelimb twitches and wake movements. For each event correlation, mean firing rate during quiescence (1–1.5 s preceding motor activity) was calculated. These means were then subtracted from each respective event correlation. For each animal, the event correlation of the pre-infusion time period was subtracted from the event correlation of the post-infusion time period. The resulting event correlations depicting the change in firing rate ($\Delta$ firing rate) were then averaged within each experimental group for each behavioral state and standard errors for every bin were calculated. Peak change in firing rate around the expected latency of sensory reafference was then calculated along with standard errors. Experimental groups within each behavioral state were compared using a paired t test. Expected latency was determined using the raw event correlations for wake (*Figure 3C*) and sleep (*Figure 3D*).

The frequencies of wake movements, twitches, and action potentials across the two time periods were calculated. Statistical significance between time periods was determined using paired t tests. For each of the measurements above, the percent difference between the pre- and post-infusion time periods was also calculated. Statistical significance between experimental groups was determined using an independent sample t test.

Unless otherwise noted, alpha was set at 0.05. Bonferroni corrections were applied when appropriate.

## Acknowledgements

This work was supported by grants from the National Institutes of Health (R37-HD081168, R01-HD063071) to MSB. We thank the members of the Blumberg lab for helpful comments on earlier drafts of the manuscript. The authors declare no competing financial interests.

## Additional information

### Funding

| Funder | Grant reference number | Author |
|---|---|---|
| National Institutes of Health | R37-HD081168 | Mark S Blumberg |
| National Institutes of Health | R01-HD063071 | Mark S Blumberg |

The funders had no role in study design, data collection and interpretation, or the decision to submit the work for publication.

### Author contributions

AT, Conception and design, Acquisition of data, Analysis and interpretation of data, Drafting or revising the article; MSB, Conception and design, Analysis and interpretation of data, Drafting or revising the article

### Author ORCIDs

Alexandre Tiriac, http://orcid.org/0000-0002-7966-981X
Mark S Blumberg, http://orcid.org/0000-0001-6969-2955

### Ethics

Animal experimentation: All experiments were carried out in accordance with the National Institutes of Health Guide for the Care and Use of Laboratory Animals (NIH Publication No. 80-23) and were approved by the Institutional Animal Care and Use Committee of the University of Iowa (protocol numbers 1202054 and 1403038).

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
