## [Decision Letter]

Congratulations: we are very pleased to inform you that your article, "Gating of reafference in the external cuneate nucleus during wake movements but not sleep-related twitches", has been accepted for publication in *eLife*. The Reviewing Editor for your submission was David Kleinfeld and the Senior Editor was Sabine Kastner.

A basic computational motif in engineering is the process of computing the difference between the intending and the actual action. Here, Tiriac and Blumberg examine forelimb movements in the rat and show that the external cuneate nucleus in the brainstem acts as a comparator of intended movement, as defined by a reference signal or so called "corollary discharge' of motor control, and the measured output from cutaneous or proprioceptive receptors or the so called "reafferent signal". In fact, if the two signals are congruent, the nucleus gates sensory input generated by the motion. Interestingly, the gate is inactive during sleep so that sensory signals from self-generated muscle twitches can activate cortical neurons.

As you will see, the reviewers have identified some minor issues that need to be fixed and made suggestions with regard to the Discussion. Please consider these issues carefully and send us an updated manuscript that we will take into production.

Reviewer #1:

This paper by Tiriac and Blumberg bears on the gating of re-afferent signals during forelimb movements generated during waking versus forelimb twitches generated during deep sleep in rat pups. It is reported that gating of re-afferent signals occurs during waking, but not during twitches, and that gating occurs in the dorsal column nuclei (the external cuneate nucleus).

This is the second report from this group on the gating of re-afferent somatosensory (proprioceptive) signals. In a prior study a gating process was documented for hindlimb movements during waking (Current Biology, 2014). What is new in the present study is that the authors provide strong evidence that gating occurs in the dorsal column nuclei. Although pre and postsynaptic inhibition has long been documented in these nuclei, no study has addressed the function of the inhibitory circuitry in somatosensory processes.

It would be interesting to determine the behavioral effect of inactivating the inhibitory circuit in dorsal column nuclei to determine whether adult rats can still differentiate between re-afferent inputs and other generated movements.

In the legend of Figure 3, panel D is not correctly identified. It should be D instead of C.

Reviewer #2:

Authors explored a critical component of their developing story implicating limb twitching during sleep as having functional significance. They investigated whether the external cuneate nucleus (ECN) acts a comparator of corollary discharge and reafferent signals in relation to wake-state forelimb movements and REM sleep-state forelimb twitches. The authors addressed this question during spontaneous waking and sleeping epochs with LFP and MU recordings in sensorimotor cortex and the ECN, with EMG recordings in forelimb, hindlimb, and nuchal muscles, and with pharmacological manipulations of the ECN. They provide compelling evidence substantiating their claims. First, they convincingly show that sensorimotor cortex responds to sleep-twitches but not to wake-movements. Second, they likewise demonstrate that the ECN has a stronger response to sleep-twitches than to wake-movements. Finally and critically, by disinhibiting ECN, the authors uncover a reafference signal in ECN in response to wake-movements without having an effect on sleep-twitches. These results represent strong evidence that the ECN meets the criteria of a comparator of corollary discharge and reafferent signals.

This is an excellent study with compelling results. Overall, I have no major concerns. The experimental design and procedures are suitable for the questions investigated, the results (and figures) are clear, and the discussion of potential mechanisms and the implications of reafferent signals during twitches in relation to development are plausible and interesting. My only recommendations are positive ones. My main suggestion is to expand upon the Discussion (last paragraph) of the potential role of twitches in organizing networks during sleep. A minor suggestion is to include the supplemental figure into the main text.

---

## [Author Response]

Reviewer #1:

This paper by Tiriac and Blumberg bears on the gating of re-afferent signals during forelimb movements generated during waking versus forelimb twitches generated during deep sleep in rat pups. It is reported that gating of re-afferent signals occurs during waking, but not during twitches, and that gating occurs in the dorsal column nuclei (the external cuneate nucleus).

[…]

Reviewer #2:

*Authors explored a critical component of their developing story implicating limb twitching during sleep as having functional significance.*

[…]

A minor suggestion is to include the supplemental figure into the main text.

We have made the following changes:

1) Fixed the error in the legend for Figure 3;

2) Moved the supplemental figure into the main text (now Figure 4); and

3) Added some language in the Discussion that expands on the possible contributions of twitches to neural network development, including the mention of spike-timing-dependent plasticity and the addition of two references.

We have also done the following:

4) Altered the title of the paper to "Gating of reafference in the external cuneate nucleus during self-generated movements in wake but not sleep"; and

5) Made several small stylistic edits to the text.